# Learning to Skim Text

## Abstract

Recurrent Neural Networks are showing much promise in many sub-areas of natural language processing, ranging from document classification to machine translation to automatic question answering. Despite their early promise, many recurrent models have to read the whole text sequentially, making it difficult to apply them to long documents. For example, it is difficult to use a recurrent network to read a book and answer questions about it. In this paper, we present an approach of reading text non-sequentially, thereby skipping irrelevant information if needed. The underlying model is a recurrent network that learns how far to jump after reading a few words of the input text. We employ a standard policy gradient method to train the model to make discrete jumping decisions. In our benchmarks on four different tasks, including number prediction, sentiment analysis, news article classification and automatic Q&A, our proposed model, a modified non-sequential LSTM, is up to 6 times faster than the standard sequential LSTM, while maintaining the same or even better accuracy.

## 1 Introduction

The last few years have seen much success of applying neural networks to many important applications in natural language processing, e.g., part-of-speech tagging, chunking, named entity recognition (Collobert et al., 2011), sentiment analysis (Socher et al., 2011, 2013), document classification (Kim, 2014; Le and Mikolov, 2014; Zhang et al., 2015; Dai and Le, 2015), machine translation (Kalchbrenner and Blunsom, 2013; Sutskever et al., 2014; Bahdanau et al., 2014; Sennrich et al., 2015; Wu et al., 2016), conversational/dialogue modeling (Sordoni et al., 2015; Vinyals and Le, 2015; Shang et al., 2015), document summarization (Rush et al., 2015; Nallapati et al., 2016), parsing (Andor et al., 2016) and automatic question answering (Q&A) (Weston et al., 2015; Hermann et al., 2015; Wang and Jiang, 2016; Wang et al., 2016; Trischler et al., 2016; Lee et al., 2016; Seo et al., 2016; Xiong et al., 2016). An important trait of all these models is that they read all the text available to them. While it is essential for certain applications, such as machine translation, this trait also makes it difficult to apply these models to scenarios that have long input text, such as document classification or automatic Q&A.

In this paper, we consider the problem of understanding long documents with partial reading, and propose a modification to the basic neural architectures that allows them to read input text non-sequentially. The main benefit of this approach is faster inference because it skips irrelevant information. An unexpected benefit of this approach is that it also helps the models generalize better.

In our approach, the model is a recurrent network, which learns to predict the number of jumping steps after it reads one or several input tokens. Such a discrete model is therefore not fully differentiable, but it can be trained by a standard policy gradient algorithm, where the reward can be the accuracy or its proxy during training.

In our experiments, we use the basic LSTM recurrent networks (Hochreiter and Schmidhuber, 1997) as the base model and benchmark the proposed algorithm on a range of document classification or reading comprehension tasks, such as Rotten Tomatoes (Pang and Lee, 2005), IMDB (Maas et al., 2011), AG News (Zhang et al., 2015) and Children's Book Test (Hill et al., 2015). We find that the proposed approach of non-sequential read-

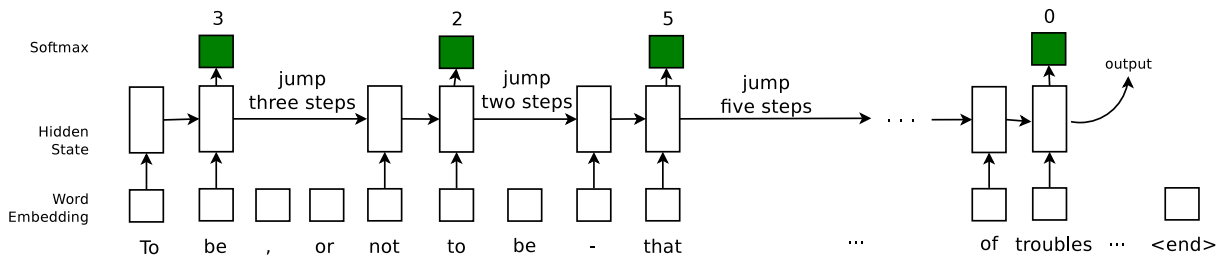

Figure 1: An example of the proposed model processing a text document. Here, the maximum size of jump is $K = 5$, the number of tokens read before a jump is $R = 2$ and the number of jumps allowed is $N = 10$. The green softmax are for jumping predictions. The processing stops if a) the jumping predicts a 0 or b) the jump times exceed $N$ or c) the network processed the last token. We only show case a) in this figure.

ing speeds up its sequential counterpart by two to six times. Surprisingly, we also observe our model beats the standard LSTM in terms of accuracy.

In summary, the main contribution of our work is to design an architecture that learns to read text non-sequentially and show that it is both faster and more accurate in practical applications of text processing. Our model is simple and flexible enough that we anticipate it be able to incorporate to recurrent nets with more sophisticated structures to achieve even better performance in the future.

## 2 Methodology

In this section, we introduce the proposed model named LSTM-Jump. We will first describe its main structure followed by the difficulty of estimating part of the model parameters because of non-differentiability. To address this issue, we appeal to a reinforcement learning formulation and adopt a policy gradient method.

### 2.1 Model Overview

The main architecture of the proposed model is shown in Figure 1, which is based on an LSTM recurrent neural network. Before training, the number of jumps $N$ allowed, the number of tokens read between every two jumps $R$ and the maximum size of jumping $K$ are chosen ahead of time. While $K$ is a fixed parameter of the model, $N$ and $R$ are hyperparameters that can vary between training and testing. We summarize those notations in Table 2, such that one can refer to when reading the experiment results in Section 3. Also, throughout the paper, we would use $d_{1:p}$ to denote a sequence $d_1, d_2, ..., d_p$.

In the following, we describe in detail how the model operates when processing text. Given a training example $x_{1:T}$, the recurrent network will read the embedding of the first $R$ tokens $x_{1:R}$ and output the hidden state. Then this state is used to compute the jumping softmax that determines a distribution over the jumping steps between 1 and $K$. The model then samples from this distribution a jumping step, which is used to decide the next token to be read into the model. Let $\kappa$ be the sampled value, then the next starting token is $x_{R+\kappa}$. Such process continues until either

 a) the jump softmax samples a 0; or

 b) the number of jumps exceeds $N$; or

 c) the model reaches the last token $x_T$.

After stopping, the latest hidden state is further used for predicting desired targets. How to leverage the hidden state depends on the specifics of the task at hand. For example, in classification problems of Section 3.1, 3.2 and 3.3, it is directly applied to produce a softmax for classification, while in automatic Q&A problem of Section 3.4, it is used to compute the correlation with the candidate answers in order to select the best one. Figure 1 gives an example with $K = 5$, $R = 2$ and $N = 10$ terminating on condition a).

### 2.2 Training with REINFORCE

Our goal for training is to estimate the LSTM and possibly word embedding parameters $\theta_m$, together with the jumping action parameters $\theta_a$. Once obtained, they can be used for inference.

The estimation of $\theta_m$ is straightforward in the tasks that can be reduced as classification problems (which is exactly what our experiments cover), as the cross entropy objective $J_1(\theta_m)$ is differentiable over $\theta_m$ that we can directly apply backpropagation to minimize.

However, the nature of discrete jumping deci-

sions made at every step makes it difficult to estimate $\theta_a$, as cross entropy is no longer differentiable over $\theta_a$. Therefore, we are tempted to formulate it as a reinforcement learning problem and apply policy gradient method to train the model. Specifically, we need to maximize a reward function over $\theta_a$ which can be constructed as follows.

Let $j_{1:N}$ be the jumping actions sequence during the training with an example $x_{1:T}$. Suppose $h_i$ is a hidden state of the LSTM right before the $i$-th jump $j_i$,[1] then it is a function of $j_{1:i-1}$ and thus can be denoted as $h_i(j_{1:i-1})$. Now the jump is attained by sampling from the multinomial distribution $p(j_i|h_i(j_{1:i-1});\theta_a)$, which is determined by the jump softmax. We can receive a reward $R$ after processing $x_{1:T}$ under the current jumping strategy.[2] The reward should be positive if the output is favorable or non-positive otherwise. In our experiments, we choose

$$R = \begin{cases} 1 & \text{if prediction correct;} \\ -1 & \text{otherwise.} \end{cases}$$

Then the objective function of $\theta_a$ we want to maximize is the expected reward under the distribution defined by the current jumping policy, i.e.,

$$J_2(\theta_a) = \mathbb{E}_{p(j_{1:N};\theta_a)}[R]. \tag{1}$$

where $p(j_{1:N};\theta_a) = \prod_i p(j_{1:i}|h_i(j_{1:i-1});\theta_a)$.

Optimizing this objective numerically requires computing its gradient, whose exact value is intractable to obtain as the expectation is over high dimensional interaction sequences. By running $S$ examples, an approximated gradient can be computed by the following REINFORCE algorithm (Williams, 1992):

$$\nabla_{\theta_a} J_2(\theta_a) = \sum_{i=1}^{N} \mathbb{E}_{p(j_{1:N};\theta_a)}[\nabla_{\theta_a} \log p(j_{1:i}|h_i;\theta_a)R]$$

$$\approx \frac{1}{S} \sum_{s=1}^{S} \sum_{i=1}^{N} [\nabla_{\theta_a} \log p(j_{1:i}^s|h_i^s;\theta_a)R^s]$$

where the superscript $s$ denotes a quantity belonging to the $s$-th example. Now the term $\nabla_{\theta_a} \log p(j_{1:i}|h_i;\theta_a)$ can be computed by standard backpropagation.

---

[1] The $i$-th jumping step is usually *not* $x_i$.

[2] In the general case, one may receive (discounted) intermediate rewards after each jump. But in our case, we only consider final reward. It is equivalent to a special case that all intermediate rewards are identical and without discount.

Although the above estimation of $\nabla_{\theta_a} J_2(\theta_a)$ is unbiased, it may have very high variance. One widely used remedy to reduce the variance is to subtract a *baseline* value $b_i^s$ from the reward $R^s$, such that the approximated gradient becomes

$$\nabla_{\theta_a} J_2(\theta_a) \approx \frac{1}{S} \sum_{s=1}^{S} \sum_{i=1}^{N} [\nabla_{\theta_a} \log p(j_{1:i}^s|h_i^s;\theta)(R^s - b_i^s)]$$

It is shown (Williams, 1992; Zaremba and Sutskever, 2015) that any number $b_i^s$ will yield an unbiased estimation. Here, we adopt the strategy of Mnih et al. (2014) that $b_i^s = w_b h_i^s + c_b$ and the parameter $\theta_b = \{w_b, c_b\}$ is learned by minimizing $(R^s - b_i^s)^2$. Now the final objective to minimize is

$$J(\theta_m, \theta_a, \theta_b) = J_1(\theta_m) - J_2(\theta_a) + \sum_{s=1}^{S} \sum_{i=1}^{N} (R^s - b_i^s)^2,$$

which is fully differentiable and can be solved by standard backpropagation.

### 2.3 Inference

During inference, we can either use sampling or greedy evaluation by selecting the most probable jumping step suggested by the jump softmax and follows that path. In the our experiments, we will adopt the sampling scheme.

## 3 Experimental Results

In this section, we present our empirical studies to understand the efficiency of the proposed model in reading text. The tasks under experimentation are: synthetic number prediction, sentiment analysis, news topic classification and automatic question answering. Those are representative tasks in text reading involving different sizes of datasets and various levels of text processing, from character to word and to sentence. Table 1 summarizes the statistics of the dataset in our experiments.

To exclude the potential impact of advanced models, we restrict our comparison between the vanilla LSTM (Hochreiter and Schmidhuber, 1997) and our model, which is referred to as LSTM-Jump. In a nutshell, we show that, while achieving the same or even better testing accuracy, our model is up to 6 times and 66 times faster than the baseline LSTM model in real and synthetic datasets, respectively, as we are able to selectively skip a large fraction of text.

In fact, the proposed model can be readily extended to other recurrent neural networks with sophisticated mechanisms such as attention and/or

| Task | Dataset | Level | Vocab | AvgLen | #train | #valid | #test | #class |
|------|---------|-------|-------|--------|--------|--------|-------|--------|
| Number Prediction | synthetic | word | 100 | 100 words | 1M | 10K | 10K | 100 |
| Sentiment Analysis | Rotten Tomatoes | word | 18,764 | 22 words | 8,835 | 1,079 | 1,030 | 2 |
| Sentiment Analysis | IMDB | word | 112,540 | 241 words | 21,143 | 3,857 | 25,000 | 2 |
| News Classification | AG | character | 70 | 200 characters | 101,851 | 18,149 | 7,600 | 4 |
| Q/A | Children Book Test-NE | sentence | 53,063 | 20 sentences | 108,719 | 2,000 | 2,500 | 10 |
| Q/A | Children Book Test-CN | sentence | 53,185 | 20 sentences | 120,769 | 2,000 | 2,500 | 10 |

Table 1: Tasks and datasets statistics.

hierarchical structure to achieve higher accuracy than those presented below. However, this is orthogonal to the main focus of this work and would be left as an interesting future work.

**General Experiment Settings** We use the Adam optimizer (Kingma and Ba, 2014) with a learning rate of 0.001 in all experiments. We also apply gradient clipping to all the trainable variables with the threshold of 1.0. The dropout rate between the LSTM layers is 0.2 and the embedding dropout rate is 0.1. We repeat the notations $N, K, R$ defined previously in Table 2, such that reader can easily refer to when looking at in Tables 4,5,6,7. While $K$ is fixed during both training and testing, we would fix $R$ and $N$ at training but vary their values during test to see how the change of parameters affects the result. Besides, the reported test time is measured by running one pass of the whole test set and the speedup is over the base LSTM model. The code is written with TensorFlow.[3]

| Notation | Meaning |
|----------|---------|
| $N$ | number of jumps allowed |
| $K$ | maximum size of jumping |
| $R$ | number of tokens read before a jump |

Table 2: Notations referred to in experiments.

## 3.1 Number Prediction with a Synthetic Dataset

First of all, we carry out the sanity check of whether LSTM-Jump is indeed able to learn how to jump if a *very clear* jumping signal is given in the text. The input of the task is a sequence of $L$ positive integers $x_{0:T-1}$ and the output is simply $x_{x_0}$. That is, the output is chosen from the input sequence, with index determined by $x_0$. Here are

---

[3] https://www.tensorflow.org/

two examples to illustrate this idea:

$$\text{input1}: \underline{4}, 5, 1, 7, \underline{6}, 2. \ \text{output1}: 6$$
$$\text{input2}: \underline{2}, 4, \underline{9}, 4, 5, 6. \ \text{output2}: 9$$

One can see that $a_0$ is essentially the indicator of how many steps the reading should jump to get the exact output and obviously, the remaining number of the sequence are useless. After reading the first token, a "smart" network should be able to learn from the training examples to jump to the output position, skipping the rest.

We generate 1 million training and 10,000 validation examples with the rule above, each with sequence length $T = 100$. We also impose $1 \leq x_0 < T$ to ensure the index is valid. We find that directly training the LSTM-Jump with full sequence is unlikely to converge, therefore, we adopt a curriculum training scheme. More specifically, we generate sequences with lengths $\{10, 20, 30, 40, 50, 60, 70, 80, 90, 100\}$ and train the model starting from the shortest. Whenever the training accuracy reaches a threshold, we shift to longer sequences. The training stops when the validation accuracy is larger than $98\%$. We also train an LSTM with the same curriculum training scheme to conduct the prediction. All the networks are single layered, with hidden size 512, embedding size 32 and batch size 100. During testing, we generate sequences of lengths 10, 100 and 1000 with the same rule, each having 10,000 examples. As the training size is huge, we do not need to worry about overfitting so dropout is not applied. In fact, we find that the training, validation and testing accuracies are almost the same.

The results of LSTM and our method, LSTM-Jump, are shown in Table 3. The first observation is that LSTM-Jump is faster than LSTM; the longer the sequence is, the more significant speed-up LSTM-Jump can gain. This is because LSTM-Jump is aware of the jumping signal at the first token and hence can directly jump to the output position to make prediction, while LSTM is agnostic to the signal and has to read the whole se-

| Seq length | LSTM-Jump | LSTM | Speedup |
|---|---|---|---|
| Test accuracy | | | |
| 10 | **98%** | 96% | n/a |
| 100 | **98%** | 96% | n/a |
| 1000 | **90%** | 80% | n/a |
| Test time (Avg tokens read) | | | |
| 10 | **13.5s (2.1)** | 18.9s (10) | 1.40x |
| 100 | **13.9s (2.2)** | 120.4s (100) | 8.66x |
| 1000 | **18.9s (3.0)** | 1250s (1000) | **66.14x** |

Table 3: Testing accuracy and time of synthetic number prediction problem. The jumping level is word.

quence. Thanks to this fact, the reading speed of LSTM-Jump is hardly affected by the length of sequence, but that of LSTM is linear with respect to length. Besides, LSTM-Jump also outperforms LSTM in terms of test accuracy under all cases. This is not surprising either, as LSTM has to read a large amount of tokens that are potentially not helpful and could interfere with the prediction. In summary, the results indicate LSTM-Jump is able to learn to "jump" when the signal is very clear.

## 3.2 Word Level Sentiment Analysis with Rotten Tomatoes and IMDB datasets

As LSTM-Jump has shown great speedups in the synthetic dataset, we would like to understand whether it could carry this benefit to real-world data, where "jumping" signal is not explicit. So in this section, we conduct sentiment analysis on two movie review datasets, both containing equal numbers of positive and negative reviews.

The first dataset is Rotten Tomatoes, which contains 10,662 documents. Since there is not a standard split, we randomly select around 80% for training, 10% for validation, and 10% for testing. The average and maximum lengths of the reviews are 22 and 56 words respectively, and we pad each of them to 60. We choose the pre-trained word2vec embeddings[4] (Mikolov et al., 2013) as our fixed word embedding that we do not update this matrix during training. Both LSTM-Jump and LSTM contain 2 layers, 256 hidden units and the batch size is 100. As the amount of training data is small, we slightly augment the data by sampling a continuous 50-word sequence in each padded reviews as one training sample. During training, we enforce LSTM-Jump to read 8 tokens before a jump ($R = 8$), and the maximum skipping to-

---

[4] https://code.google.com/archive/p/word2vec/

kens per jump is 10 ($K = 10$), while the number of jumps allowed is 3 ($N = 3$).

The testing result is reported in Table 4. In a nutshell, LSTM-Jump is always faster than LSTM under different combinations of $R$ and $N$. At the same time, the accuracy is on par with that of LSTM. In particular, the combination of $(R, N) = (7, 4)$ even achieves slightly better accuracy than LSTM while having a 1.5x speedup.

| Model | $(R, N)$ | Accuracy | Time | Speedup |
|---|---|---|---|---|
| LSTM-Jump | (9, 2) | 0.783 | **6.3s** | **1.98x** |
| | (8, 3) | 0.789 | 7.3s | 1.71x |
| | (7, 4) | **0.793** | 8.1s | 1.54x |
| LSTM | n/a | 0.791 | 12.5s | 1x |

Table 4: Testing time and accuracy on the Rotten Tomatoes review classification dataset. The maximum size of jumping $K$ is set to 10 for all the settings. The jumping level is word.

The second dataset of interest is IMDB (Maas et al., 2011),[5] which contains 25,000 training and 25,000 testing movie reviews, where the average length of text is 240 words, much longer than that of Rotten Tomatoes. We randomly set aside about 15% of training data as validation set. Both LSTM-Jump and LSTM has one layer and 128 hidden units, and the batch size is 50. Again, we use pretrained word2vec embeddings as initialization but they are updated during training. We either pad a short sequence to 400 words or randomly select a 400-word segment from a long sequence as a training example. The number of tokens read before a jump is set to $R = 20$, maximum skipping tokens per jump is $K = 40$ and the maximum number of jumps is $N = 5$.

| Model | $(R, N)$ | Accuracy | Time | Speedup |
|---|---|---|---|---|
| LSTM-Jump | (80, 8) | **0.894** | 769s | 1.62x |
| | (80, 3) | 0.892 | 764s | 1.63x |
| | (70, 3) | 0.889 | 673s | 1.85x |
| | (50, 2) | 0.887 | 585s | 2.12x |
| | (100, 1) | 0.880 | **489s** | **2.54x** |
| LSTM | n/a | 0.891 | 1243s | 1x |

Table 5: Testing time and accuracy on the IMDB sentiment analysis dataset. The maximum size of jumping $K$ is set to 40 for all the settings. The jumping level is word.

As Table 5 shows, the result exhibits a similar trend as found in Rotten Tomatoes that LSTM-

---

[5] http://ai.Stanford.edu/amaas/data/sentiment/index.html

Jump is uniformly faster than LSTM under many settings. The various $(R, N)$ combinations again display the trade-off between efficiency and accuracy. If one cares more about accuracy, then allowing LSTM-Jump to read and jump more times is a good choice. Otherwise, shrinking either one would bring a significant speedup though at the price of losing some accuracy. Nevertheless, the configuration with the highest accuracy still enjoys a 1.6x speedup compared to LSTM. With a slight loss of accuracy, LSTM-Jump can be 2.5x faster .

### 3.3 Character Level News Article Classification with AG dataset

We now present results on testing the character level jumping with a news article classification problem. The data contains four classes of topics (World, Sports, Business, Sci/Tech) from the AG's news corpus,[6] a collection of more than 1 million news articles. The data we use is the subset constructed by Zhang et al. (2015) for classification with character-level convolutional networks. There are 30,000 training and 1900 testing examples for each class respectively, where 15% of training are set aside as validation. The non-space alphabet under use are:

```
abcdefghijklmnopqrstuvwxyz0123456
789-,;.!?:/\|_@#$%&* +-=<>()[]{}
```

Since the vocabulary size is small, we choose 16 as the embedding size. The initialized entries of the embedding matrix are drawn from a uniform distribution in $[-0.25, 0.25]$, which are progressively updated during training. Both LSTM-Jump and LSTM have 1 layer and 64 hidden units and the batch sizes are 20 and 100 respectively. The training sequence is again of length 400 that it is either padded from a short sequence or sampled from a long one. The number of characters read before a jump is $R = 30$, the maximum span per jump is $K = 40$ and $N = 5$ jumps are allowed at training.

The result is summarized in Table 6. Not surprisingly, LSTM-Jump outperforms LSTM in terms of both efficiency and accuracy, although the advantage in speedup is not as significant as that in the previous tasks. This is mainly due to the fact that the embedding size and hidden are both much smaller than those used previously, and accordingly the processing of a token is much faster. In that case, other computation overhead such as

calculating and sampling from the jump softmax might become a dominating factor of efficiency. By this cross-tasks comparison, we can see that the larger the recurrent neural network and the embedding are, the more speedup LSTM-Jump can gain, which is also confirmed by the task below.

| Model | $(R, N)$ | Accuracy | Time | Speedup |
|---|---|---|---|---|
| LSTM-Jump | (50, 5) | 0.854 | 102s | 0.80x |
| | (40, 6) | 0.874 | 98.1s | 0.83x |
| | (40, 5) | 0.889 | 83.0s | 0.98x |
| | (30, 5) | 0.885 | **63.6s** | **1.28x** |
| | (30, 6) | **0.893** | 74.2s | 1.10x |
| LSTM | n/a | 0.881 | 81.7s | 1x |

Table 6: Testing time and accuracy on the AG news classification dataset. The maximum size of jumping $K$ is set to 40 for all the settings. The jumping level is character.

### 3.4 Sentence Level Automatic Question Answering with Children's Book Test dataset

The last task is automatic question answering, in which we aim to test the sentence level skimming of LSTM-Jump. We benchmark on the data set Children's Book Test (CBT) (Hill et al., 2015).[7] In each document, there are 20 contiguous sentences (context) extracted from a children's book followed by a query sentence. A word of the query is deleted and the task is to select the best fit for this position from 10 candidates. Originally, there are 4 types of tasks according to the part of speech of the missing word, from which, we choose the most difficult two, i.e., the name entity (NE) and common noun (CN) as our focus, since simple language models can already achieve human-level performance for the rest two types .

The models, LSTM or LSTM-Jump, firstly read the whole query, then the context sentences and finally output the predicted word. While LSTM reads everything, our jumping model would decide how many context sentences should skip after reading one sentence. Whenever a model finishes reading, the context and query are encoded in its hidden state $h_O$, and the best answer from the candidate words has the same index that maximizes the following:

$$\text{softmax}(CWh_O) \in \mathbb{R}^{10},$$

---

[6] http://www.di.unipi.it/~gulli/AG_corpus_of_news_articles.html

[7] http://www.thespermwhale.com/jaseweston/babi/CBTest.tgz

where $C \in \mathbb{R}^{10 \times d}$ is the word embedding matrix of the 10 candidates and $W \in \mathbb{R}^{d \times \text{hidden\_size}}$ is a trainable weight variable. Using such bilinear form to select answer basically follows the idea of Chen et al. (2016), as it is shown to have good performance. The task is now distilled to a classification problem of 10 classes.

We either truncate or pad each context sentence, such that they all have length 20. The same pre-processing is applied to the query sentences except that the length is set 30. For both models, the number of layers is 2, the hidden units are 256 and the batch size is 32. Pretrained word2vec embeddings are again used and they are not adjusted during training. The maximum number of context sentences LSTM-Jump can skip per time is $K = 5$ while the number of total jumping is limited to $N = 5$. We let the model jump after reading every sentence, so $R = 1$ (20 words).

The result is reported in Table 7. The performance of LSTM-Jump is superior to LSTM in terms of both accuracy and efficiency under all settings in our experiments. In particular, the fastest LSTM-Jump configuration achieves a remarkable 6x speedup over LSTM, while also having respectively 1.4% and 4.4% higher accuracy in Children's Book Test - Named Entity and Children's Book Test - Common Noun.

| Model | $(R, N)$ | Accuracy | Time | Speedup |
|---|---|---|---|---|
| Children's Book Test - Named Entity | | | | |
| LSTM-Jump | (1, 5) | **0.468** | 40.9s | 3.04x |
| | (1, 3) | 0.464 | 30.3s | 4.11x |
| | (1, 1) | 0.452 | **19.9s** | **6.26x** |
| LSTM | n/a | 0.438 | 124.5s | n/a |
| Children's Book Test - Common Noun | | | | |
| LSTM-Jump | (1, 5) | 0.493 | 39.3s | 3.09x |
| | (1, 3) | 0.487 | 29.7s | 4.09 |
| | (1, 1) | **0.497** | **19.8s** | **6.14x** |
| LSTM | n/a | 0.453 | 121.5s | 1x |

Table 7: Testing time and accuracy on the Children's Book Test dataset. The maximum size of jumping $K$ is set to 5 for all the settings. The jumping level is sentence.

The dominant performance of LSTM-Jump over LSTM might be interpreted as follows. After reading the query, both LSTM and LSTM-Jump know what the question is. However, LSTM still has to process the remaining 20 sentences and thus at the very end of the last sentence, the long dependency between the question and output might become weak that the prediction is hampered. On the contrary, the question can guide LSTM-Jump

on how to read selectively and stop early when the answer is clear. Therefore, when it comes to the output stage, the "memory" is both fresh and uncluttered that a more accurate answer is likely to be picked.

Below is an example of how the model reads a test context given a query (bold face sentences are those read by our model in the increasing order). XXXXX is the missing word we want to fill.

(a) *Query:* 'XXXXX!

(b) *Context:*
1. **said Big Klaus, and he ran off at once to Little Klaus.**
2. 'Where did you get so much money from?'
3. **'Oh, that was from my horse-skin.**
4. I sold it yesterday evening.'
5. 'That 's certainly a good price!'
6. said Big Klaus; and running home in great haste, he took an axe, knocked all his four
7. **'Skins!**
8. **skins!**
9. Who will buy skins?'
10. he cried through the streets.
11. All the shoemakers and tanners came running to ask him what he wanted for them.'
12. **A bushel of money for each,' said Big Klaus.**
13. 'Are you mad?'
14. **they all exclaimed.**
15. 'Do you think we have money by the bushel?'
16. 'Skins!
17. skins!
18. Who will buy skins?'
19. he cried again, and to all who asked him what they cost, he answered,' A bushel
20. 'He is making game of us,' they said; and the shoemakers seized their yard measures and

(c) *Candidates:* Klaus | Skins | game | haste | head | home | horses | money | price | streets

(d) *Answer:* Skins

The reading behavior might be interpreted as follows. The model tries to search for clues, and after reading sentence 8, it realizes that the most plausible answer is "Klaus" or "Skins", as they both appear twice. "Skins" is more likely to be the answer as it is followed by a "!". The model searches further to see if "Klaus!" is mentioned somewhere, but it only finds "Klaus" without "!" for the third time. After the last attempt at sentence 14, it is confident about the answer and stops to output with "Skins".

## 4   Related Work

Closely related to our work is the idea of learning visual attention with neural networks (Mnih et al., 2014; Ba et al., 2014; Sermanet et al., 2014), where a recurrent model is used to combine visual evidence at multiple fixations processed by a convolutional neural network. Similar to our approach, the model is trained end-to-end using the REINFORCE algorithm (Williams, 1992). However, a major difference between those work and ours is that we have to sample from discrete jumping distribution, while they can sample from continuous distribution such as Gaussian. The difference is mainly due to the inborn characteristics of text and image. In fact, as pointed out by Mnih et al. (2014), it was difficult to learn policies over more than 25 possible discrete locations.

This idea has recently been explored in the context of natural language processing applications, where the main goal is to filter irrelevant content using a small network (Choi et al., 2016). Perhaps most closely related to our work is the concurrent work on learning to reason with reinforcement learning (Shen et al., 2016). The key difference between our work and Shen et al. (2016) is that they focus on early stopping after multiple pass of data to ensure accuracy whereas our method focuses on selective reading with single pass to enable fast processing.

The concept of "hard" attention has also used successfully in the context of making neural network predictions more interpretable (Lei et al., 2016). The key difference between our work and Lei et al. (2016)'s method is that our method optimizes for faster inference, and is more dynamic in its jumping. Likewise is the difference between our approach and the "soft" attention approach by (Bahdanau et al., 2014).

Our method belongs to adaptive computation of neural networks, whose idea is recently explored by (Graves, 2016; Jernite et al., 2016), where different amount of computations are allocated dynamically per time step. The main difference between our method and Graves; Jernite et al.'s methods is that our method can set the amount of computation to be exactly zero for many steps, thereby achieving faster scanning over texts. Even though our method requires policy gradient methods to train, which is a disadvantage compared to (Graves, 2016; Jernite et al., 2016), we do not find training with policy gradient methods problematic in our experiments.

At the high-level, our model can be viewed as a simplified trainable Turing machine, where the controller can move on the input tape. It is therefore related to the prior work on Neural Turing Machines (Graves et al., 2014) and especially its RL version (Zaremba and Sutskever, 2015). Compared to (Zaremba and Sutskever, 2015), the output tape in our method is more simple and reward signals in our problems are less sparse, which explains why our model is easy to train. It is worth noting that Zaremba and Sutskever report difficulty in using policy gradients to train their model.

Our method, by skipping irrelevant content, shortens the length of recurrent networks, thereby addressing the vanishing or exploding gradients in them (Hochreiter et al., 2001). The baseline method itself, Long Short Term Memory (Hochreiter and Schmidhuber, 1997), belongs to the same category of methods. In this category, there are several recent methods that try to achieve the same goal, such as having recurrent networks that operate in different frequency (Koutnik et al., 2014) or is organized in a hierarchical fashion (Chan et al., 2015; Chung et al., 2016).

## 5   Conclusion and Future Work

In this paper, we focus on learning how to skim text for fast reading. In particular, we propose a "jumping" model that after reading every few tokens, it decides how many tokens should be skipped by sampling from a softmax. Such jumping behavior is modeled as a discrete decision making process, which can be trained by reinforcement learning algorithm such as REINFORCE. In four different tasks with six datasets, we test the efficiency of the proposed method on various levels of text jumping, from character to to word and to sentence. The results indicate our model is several times faster than, while the accuracy is on par with the baseline LSTM model.

As an important future work, we hope to extend our model to a bidirectional jumping network, such that it can jump back and forth and possibly pay more attention to the important part of text. We would also like to incorporate our model with advanced neural networks such as memory network (Weston et al., 2014), and/or with sophisticated mechanisms like attention (Bahdanau et al., 2014), and/or with hierarchical structure (Choi et al., 2016) to build a more accurate model.

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
