# Peer review of "Learning to Skim Text"

_ACL 2017 — decision unknown_

[Official Review · Reviewer 1 · rating 4 · confidence 4]
soundness 3 · originality 4 · clarity 5 · impact 4 · substance 4 · appropriateness 5 · meaningful comparison 5 · presentation format Oral Presentation

The paper proposes a recurrent neural architecture that can skip irrelevant
input units. This is achieved by specifying R (# of words to read at each
"skim"), K (max jump size), and N (max # of jumps allowed). An LSTM processes R
words, predicts the jump size k in {0, 1...K} (0 signals stop), skips the next
k-1 words and continues until either the number of jumps reaches N or the model
reaches the last word. While the model is not differentiable, it can be trained
by standard policy gradient. The work seems to have been heavily influenced by
Shen et al. (2016) who apply a similar reinforcement learning approach
(including the same variance stabilization) to multi-pass machine reading. 

- Strengths:

The work simulates an intuitive "skimming" behavior of a reader, mirroring Shen
et al. who simulate (self-terminated) repeated reading. A major attribute of
this work is its simplicity. Despite the simplicity, the approach yields
favorable results. In particular, the authors show through a well-designed
synthetic experiment that the model is indeed able to learn to skip when given
oracle jump signals. In text classification using real-world datasets, the
model is able to perform competitively with the non-skimming model while being
clearly faster. 

The proposed model can potentially have meaningful practical implications: for
tasks in which skimming suffices (e.g., sentiment classification), it suggests
that we can obtain equivalent results without consuming all data in a
completely automated fashion. To my knowledge this is a novel finding. 

- Weaknesses:

It's a bit mysterious on what basis the model determines its jumping behavior
so effectively (other than the synthetic dataset). I'm thinking of a case where
the last part of the given sentence is a crucial evidence, for instance: 

"The movie was so so and boring to the last minute but then its ending blew me
away." 

In this example, the model may decide to skip the rest of the sentence after
reading "so so and boring". But by doing so it'll miss the turning point
"ending blew me away" and mislabel the instance as negative. For such cases a
solution can be running the skimming model in both directions as the authors
suggest as future work. But in general the model may require more sophisticated
architecture for controlling skimming.

It seems one can achieve improved skimming by combining it with multi-pass
reading (presumably in reverse directions). That's how humans read to
understand text that can't be digested in one skim; indeed, that's how I read
this draft. 

Overall, the work raises an interesting problem and provides an effective but
intuitive solution.